# Liver Extracellular Matrix in Colorectal Liver Metastasis

**DOI:** 10.3390/cancers17060953

**Published:** 2025-03-12

**Authors:** Marika Morabito, Pauline Thibodot, Anthony Gigandet, Philippe Compagnon, Christian Toso, Ekaterine Berishvili, Stéphanie Lacotte, Andrea Peloso

**Affiliations:** 1General, Emergency and Transplant Surgery Department, ASST Settelaghi, University Hospital and Faculty of Medicine of Insubria, 21100 Varese, Italy; 2Hepato-Biliary Center, Paul-Brousse Hospital, Assistance Publique-Hôpitaux de Paris, 94800 Villejuif, France; 3School of Medecine, Faculty of Medecine, University of Geneva, 1211 Geneva, Switzerland; 4Division of Transplantation, Department of Surgery, Geneva University Hospitals and Faculty of Medicine, 1205 Geneva, Switzerland; philippe.compagnon@hcuge.ch; 5Division of Abdominal Surgery and Transplantation, Department of Surgery, Geneva University Hospitals and Faculty of Medicine, 1205 Geneva, Switzerland; 6Cell Isolation and Transplantation Center, Department of Surgery, Geneva University Hospitals and University of Geneva, 1211 Geneva, Switzerland; ekaterine.berishvili@unige.ch; 7Hepatology and Transplantation Laboratory, Department of Surgery, Faculty of Medicine, University of Geneva, 1206 Geneva, Switzerland; stephanie.lacotte@unige.ch

**Keywords:** ECM, CRLM, metastatic niche, chemoresistance

## Abstract

The liver is the most common site of metastasis from colorectal cancer (CRC) metastases, a major cause of cancer-related deaths. This review highlights the pivotal role of the hepatic extracellular matrix (ECM) in supporting tumor growth, invasion, and chemoresistance in colorectal liver metastases (CRLM). The liver’s ECM, composed of collagens, fibronectin, laminins, and proteoglycans, undergoes significant remodeling during metastasis. ECM stiffening and collagen cross-linking, driven by matrix metalloproteinases (MMPs) and lysyl oxidases (LOX), create a microenvironment that promotes cancer cell adhesion, invasion, angiogenesis, and immune evasion. ECM remodeling is also linked to chemoresistance. Elevated ECM components activate survival pathways like FAK and PI3K/AKT, suppressing apoptosis and limiting drug efficacy. Emerging therapies targeting ECM components, including MMP and LOX inhibitors, show promise for disrupting metastasis and overcoming chemoresistance, underscoring the need for further research into ECM-targeted strategies for CRLM treatment.

## 1. Introduction

Colorectal cancer (CRC) is recognized as one of the most burdensome malignancies worldwide, ranks as the third most frequent neoplastic disease globally, and stands as the second leading cause of cancer-related mortality, with over 1.9 million new cases and ≈935,000 deaths every year [1]. According to recent estimates, there will be approximately 2,001,140 new cancer cases and 611,720 cancer deaths in the United States in 2024, reflecting the significant public health impact of cancer overall [2]. Its incidence differs by region, and rates are higher in developed countries (which is determined by lifestyle, diet, and surveillance measures [3]). Although CRC tends to occur in people over 50 years of age, there is an increasing incidence of the disease in young people. The incidence is slightly higher in men than in women, and this difference is attributed to lifestyle and genetics [4]. Depending on the stage, the five-year survival rate for patients with localized CRC is approximately 90%, whereas, notably, it plummets dramatically to roughly 14% once the disease advances to the metastatic stages [5,6]. This underscores the critical need for a deeper understanding of the mechanisms that drive metastatic dissemination, particularly to the liver, which remains the most common target for CRC metastasis [7] with up to 25% of patients already found to have synchronous liver metastases [8]. Furthermore, an additional 25% of patients will develop liver metastasis within the subsequent three years [9] after resection of the primary tumor.

Over the past several decades, the landscape of cancer therapy has evolved substantially, from radical surgical procedures and conventional chemotherapy to the advent of targeted therapies and immunotherapies. Advances in molecular biology and genomic profiling have led to the development of more personalized treatment regimens, which have improved outcomes in various malignancies [10]. Nevertheless, challenges such as therapeutic resistance and treatment-related toxicities persist, reinforcing the urgency of continued research and innovation. In this context, identifying robust prognostic biomarkers is essential not only to refine therapeutic approaches but also to inform clinical decision-making and health policy, ultimately aiming to reduce the overall cancer burden.

The liver’s unique susceptibility to CRC metastasis is largely attributed to the fact that the majority of intestinal mesenteric venous drainage is directed into the hepatic portal venous system. This anatomical arrangement essentially delivers tumor cells directly from the colon to the liver [11]. Although the hematogenous route via the portal venous system is the predominant mechanism for liver metastasis, other pathways also contribute to the complex process of metastatic dissemination. CRC cells can spread via the lymphatic system, though this is a less common route of liver involvement, potentially contributing to secondary liver metastasis following lymphatic dissemination [11,12,13,14]. In advanced disease, CRC may also invade the peritoneal cavity, with subsequent direct extension into the liver [15,16].

Despite extensive research into CRC and its metastatic behavior, the biological mechanisms underlying the spread of cancer from the colon to the liver remain incompletely understood, particularly concerning the interaction between disseminating tumor cells and the hepatic microenvironment.

Several mechanisms have been identified facilitating CRC metastasis [Figure 1]:(1)CRC cells may undergo epithelial-mesenchymal transition (EMT), a process in which they lose their epithelial traits and acquire mesenchymal properties, thereby enhancing their ability to migrate and invade distant tissues [17]: this is principally driven by genetic mutations (such as KRAS, TP53, and APC) [18] and epigenetic alterations (like DNA methylation) [19] which influence CRC metastatic potential;(2)CRC cells can enter the bloodstream as circulating tumor cells (CTCs), where they must survive the challenges of the circulatory environment, extravasate into the liver parenchyma, and subsequently establish metastatic colonies [20];(3)CRC metastasis aligns with Paget’s ’Seed and Soil Hypothesis’ proposed in 1889 [21], suggesting that the liver provides a favorable microenvironment (“soil”) that enables CRC cells (“seeds”) to survive and proliferate, facilitated by its rich blood supply and specific extracellular matrix (ECM) components [22].

However, it is still unclear how CRC cells interact with hepatic ECM and more in general with the so-called liver microenvironment to allow the cells to survive, home, and colonize, and what steps and critical factors govern this process. Addressing these gaps is crucial for improving outcomes in patients with metastatic CRC.

This review aims to comprehensively assess the role of hepatic ECM in colorectal liver metastasis (CRLM). We will also discuss how liver ECM changes affect the behavior of metastatic CRC cells and their resistance to chemotherapy. The focus will be on the ECM of the liver in CRC metastasis, describing how its unique pattern of composition and remodeling favors the metastatic CRC cells. Through the integration of clinical and basic research, we aim to identify potential therapeutic targets within the ECM and develop strategies to improve patient outcomes.

Schematic of the metastatic cascade in colorectal cancer (CRC), focusing on liver metastases. The process involves (1) local invasion and intravasation, (2) circulation as circulating tumor cells (CTCs), and (3) extravasation and colonization in the liver. Key concepts include the “seed and soil” theory, epithelial-to-mesenchymal transition (EMT), genetic alterations (e.g., KRAS, TP53, and APC), and epigenetic modifications (e.g., DNA methylation). The right panel highlights the tumor microenvironment, including stromal interactions, angiogenesis, and immune modulation, essential for metastatic growth.

## 2. The Extracellular Matrix: Composition and Function in Healthy Tissue

The ECM is a complex, dynamic network of over 300 molecules that provides scaffolding to maintain tissue structure and integrity while also enabling cells to organize into functional tissues. The ECM’s unique composition and mechanical properties determine the physical characteristics of each tissue, such as its elasticity, strength, and rigidity [23]. Beyond structural support, the ECM is a critical player in cell signaling: it influences numerous cellular processes, such as proliferation, differentiation, migration, and apoptosis [24]. This signaling occurs through interactions between ECM components and cell surface receptors, predominantly integrins [25]. Additionally, the ECM mediates the transmission of mechanical signals to cells (mechanotransduction), which is critical for maintaining tissue architecture and function.

Its major components can be divided into structural proteins (such as collagens and elastins, which provide tensile strength and elasticity) [26,27,28,29], adhesive glycoproteins (including fibronectins and laminins, which mediate cell–ECM interactions) [26], and proteoglycans and glycosaminoglycans (GAGs) [30,31].

### 2.1. Collagens

Collagens represent the most abundant protein family within the ECM, providing structural integrity and mechanical stability to tissues [32]. There are at least 28 types of collagens described, classified into fibrillar and non-fibrillar groups, each playing specific roles in different tissue types. Elastins are essential for tissue elasticity, particularly abundant in tissues that require stretching and recoiling, such as the lungs, skin, blood vessels, and liver, which undergo constant expansion and contraction due to physiological action [33].

### 2.2. Glycosaminoglycans (GAGs)

Glycosaminoglycans (GAGs) are long, unbranched polysaccharides consisting of repeating disaccharide units that contribute to the ECM’s hydration and mechanical properties by attracting water molecules [34,35]. Common GAGs include hyaluronic acid, chondroitin sulfate, heparan sulfate, and keratan sulfate [36]. Hyaluronic acid (HA) is the most abundant non-sulfated GAG in the ECM, known for its ability to retain water and create a hydrated environment conducive to cellular movement. Heparan sulfate (HS) is another critical GAG that interacts with growth factors, cytokines, and ECM proteins to regulate signaling pathways involved in cell growth and survival [37]. Proteoglycans are proteins that are made up of a core covalently linked to one or more GAG chains. Less representative GAGs include decorin [38], aggrecan (primarily known for its role in cartilage but also involved in maintaining tissue hydration and compressive strength) [39], and versican [40].

### 2.3. Glycoproteins

Among the ECM glycoproteins, fibronectin is the most represented, with more than 20 different isoforms. Assembled into fibers by cells [41], each fibronectin molecule consists of three modules of repeated motifs with distinct structures [42]. These modules contain binding motifs that enable fibronectin molecules to interact with each other as well as cellular receptors, collagen, and gelatin. It is found as a dimer at the C-terminal, linked by disulfide bridges. The binding of these dimers to cellular integrins leads to the activation and clustering of integrins, which promotes intermolecular interactions of fibronectin and the formation of fibrils.

### 2.4. Metalloproteinases (MMPs) and Tissue Inhibitors of Metalloproteinase (TIMPs)

Matrix metalloproteinases (MMPs) are endopeptidases involved in tissue remodeling through selective proteolytic degradation of the ECM components, and play a crucial role in maintaining tissue homeostasis [43,44]. In healthy tissues, a precise balance between MMPs and their natural inhibitors, tissue inhibitors of metalloproteinases (TIMPs), is maintained to regulate ECM turnover and ensure proper tissue function. This MMP/TIMP equilibrium is essential for various physiological processes, such as wound healing and tissue regeneration, preventing uncontrolled matrix degradation that could otherwise lead to tissue damage. Disruption of this balance, with elevated MMP activity, can result in pathological states characterized by excessive tissue remodeling and functional impairment [45].

## 3. Extracellular Matrix (ECM) in Cancer

The ECM is crucial not only in supporting tumor cell growth but also in driving malignant transformation. Within primary tumor tissue, the ECM undergoes substantial remodeling, which contributes to an altered microenvironment that promotes tumor cell proliferation and survival [46]. The tumoral microenvironment is a highly complex and heterogeneous ambiance, composed of cellular elements (fibroblasts, endothelial cells, adipocytes, and immune and inflammatory cells) and non-cellular components, primarily the ECM [47]. Key proteins such as fibronectin, collagen, and matrix metalloproteinases (MMPs) are frequently upregulated, enabling cancer cells to circumvent the constraints of normal growth regulation [48]. These ECM alterations facilitate the tumor’s capacity for uncontrolled expansion and resistance to standard cellular checkpoints.

During the metastatic process, the ECM helps cancer progression by modifying adhesion properties and degrading the surrounding tissues, thereby promoting invasion into adjacent structures, such as blood vessels, and then entering into the bloodstream. Once tumor cells enter the bloodstream, the ECM components assist in the establishment of secondary tumors by providing a supportive niche for metastatic colonization [49].

Recent research underscores how cancer cells exploit ECM remodeling and interactions during cancer metastasis particularly in:Pre-metastatic niche formation;Metastatic niche colonization;Movement of circulating tumoral cells (CTCs).

### 3.1. ECM Role in Pre-Metastatic Niche Formation

The formation of the pre-metastatic niche, intended as a favorable microenvironment in distant organs that supports the colonization of circulating tumor cells from the primary tumor, is a pivotal early event in cancer metastasis [50,51]. ECM remodeling is central to this process through cellular and non-cellular components [52,53]. During cancer progression, carcinoma cells recruit stromal cells, altering their properties and metabolism. Together, they also reshape the ECM via enzymes such as MMPs and lysyloxidases (LOX), which degrade and cross-link ECM proteins [54]. This remodeling leads to a typical increased stiffness and altered ECM composition, which promotes tumor invasion by creating a more supportive environment for cancer cell migration and growth. Tumor cells also release extracellular vesicles (EVs) containing nucleic acids, lipids, and proteins, which further contribute to ECM remodeling, fibroblast activation, angiogenesis, immune modulation, and the formation of pre-metastatic niches [55]. In the context of pre-metastatic niche formation, fibronectin, tenascin-C, and collagen are often upregulated in distant tissues, providing critical structural and biochemical signals that prepare these sites for metastatic colonization [56]. Proteolytic enzymes, like LOX carried by tumor-derived EVs, cross-link collagen fibers in these niches, resulting in a stiffened matrix that facilitates the adhesion of circulating tumor cells. In addition to structural changes, the ECM interacts with growth factors and cytokines, further amplifying pro-metastatic signaling pathways, thereby enhancing the metastatic potential of cancer cells [57]. Additionally, even ECM degradation is not just a simple, passive event; it releases matrix-bound growth factors and matrikines (including Laminin-5 γ2 Chain Peptide or Laminin-1 α1 Chain Peptide), which engage with surface receptors to trigger signal transduction pathways that regulate tumor growth, cell migration, and invasion. This dynamic interplay between ECM remodeling and cellular signaling is central to the establishment of a pro-tumorigenic microenvironment, both at the primary tumor site and within distant pre-metastatic niches [58]. By creating a permissive environment that supports angiogenesis, inflammation, and immune cell recruitment, the ECM actively participates in shaping the microenvironment required for tumor progression and metastasis.

### 3.2. Cell–ECM Interactions in the Metastatic Niche

Once cancer cells reach a metastatic site, their survival, proliferation, and resistance to apoptosis are strongly shaped by their interactions with the local ECM [59]. These interactions primarily involve integrins and other ECM receptors, which activate key signaling pathways, such as FAK and PI3K/Akt. These pathways support cancer cell survival and protect them from anoikis, a form of apoptosis triggered by detachment from the ECM [60]. Additionally, cancer-associated fibroblasts (CAFs) and other stromal cells further remodel the ECM at the metastatic site. This remodeling increases the number of available adhesion sites and amplifies growth-promoting signals, allowing metastatic cells to establish themselves and form secondary tumors [61]. Crucially, certain ECM components, like hyaluronan and laminin, provide essential cues that guide the migration and invasion of metastatic cells through their new microenvironment, facilitating tumor progression and colonization [62].

### 3.3. Circulating Tumoral Cells (CTCs) and ECM Interactions

For metastasis to occur, cancer cells must first detach from the primary tumor, enter the bloodstream as circulating tumor cells (CTCs), and then extravasate into distant tissues [63]. The ECM plays a pivotal role in each of these stages. The detachment of cancer cells from the primary tumor is often facilitated by ECM degradation, largely driven by MMPs, which break down ECM proteins. This degradation enables cancer cells to invade surrounding tissues and gain access to the vasculature [64,65]. Once in the bloodstream, CTCs must adhere to the endothelial lining of distant organs to begin the extravasation process. ECM proteins, such as fibronectin and collagen, present on the endothelial surface, promote CTC adhesion via integrins, initiating their exit from circulation. After adhering, CTCs employ ECM-remodeling enzymes, including heparinase [66,67] and MMPs [68], to degrade the endothelial basement membrane [69], allowing them to infiltrate the underlying tissue and establish new colonies. The ECM’s role in metastasis underscores the intricate relationship between cancer cells and their surrounding microenvironment. Far from being a static structure, the ECM actively modulates tumor progression by supporting key metastatic processes. Understanding the molecular mechanisms by which the ECM influences metastasis presents potential therapeutic opportunities to inhibit cancer spread.

These findings underscore the intricate interplay between ECM components and tumor behavior, setting the stage for exploring the emerging role of cRNA in modulating ECM remodeling and cancer progression.

Circular RNAs (circRNAs) play pivotal roles in CRLM, influencing tumor progression through modulation of ECM dynamics. These non-coding RNAs, defined by their stable closed-loop structures, regulate key metastatic processes such as tumor cell adhesion, invasion, and apoptosis through various molecular interactions [22,70]. Recent studies highlight the involvement of specific circRNAs, including circ_0017552, which modulates colon cancer proliferation and apoptosis by functioning as a sponge for miR-338–3p, consequently upregulating NET1 expression [71]. NET1, a known oncogene, enhances tumor proliferation, invasion, and anti-apoptotic capabilities, thus promoting metastatic progression. Furthermore, circ_0017552 expression itself is transcriptionally induced by the transcription factor SP1, adding an additional layer of complexity to circRNA regulatory mechanisms in colorectal cancer [71]. Importantly, circRNAs also influence the ECM by altering the expression of MMPs, integrins, and other ECM-associated proteins, thereby significantly reshaping the tumor microenvironment and facilitating metastatic colonization in the liver [22]. Thus, dissecting circRNA-driven modulation of ECM dynamics represents a promising strategy for identifying novel biomarkers and therapeutic targets to impede colorectal cancer liver metastasis.

## 4. Liver ECM in Regulating Cell Fate, and Its Role in CRC Metastasis

### 4.1. Unique Characteristics of the Hepatic ECM

The liver ECM is a specialized, dynamic network of macromolecules crucial for maintaining hepatic structure and function. In a healthy liver, the ECM is generally sparse: the liver’s ECM differs from other tissues due to its unique composition and organization, tailored to support the liver’s metabolic, detoxification, and regenerative functions [72]. It is physiologically organized into distinct zones corresponding to its functional architecture, including the periportal, midzonal, and perivenous regions. This zonal heterogeneity reflects the liver’s diverse metabolic activities and cellular composition as well as ECM differentiation [73] with bi-directional pathways. The sinusoidal endothelial cells are characterized by an incomplete basement membrane, which facilitates the efficient exchange of materials. Nonetheless, key fibrillar ECM proteins such as laminin, collagen IV, and fibronectin are present along the sinusoids [74,75]. Recent proteomic profiling of the matrisome in a healthy liver has identified a more complex variety of core ECM components, including collagens, fibulins, annexins, and elastins [76,77,78]. Among these, collagens I, IV, and VI are notably abundant. Collagens IV and VI are integral to the liver’s BM structures, while collagen I predominantly resides within the interstitial ECM [Table 1]. Three-dimensional imaging studies of the interstitial collagens, carried out using decellularized liver preparations, have illustrated unique structural arrangements for collagens I, VII, and XIV, revealing diverse patterns of fiber bundling and alignment, which significantly impact tissue mechanics [79]. It is noteworthy that hepatocytes and cholangiocytes, rather than hepatic stellate cells (HSCs), secrete these ECM proteins in a healthy liver. Nevertheless, single-cell RNA sequencing (RNAseq) of murine HSCs has demonstrated distinct matrisome component expression patterns along the centroportal axis. Specifically, Podn, Loxl1, and Adamtsl2 exhibit pericentral expression, while Igfbp3 and Itgb3 are expressed in periportal regions [80]. Although the functional implications of these spatial variations remain unclear, they warrant additional exploration [81].

In the liver and metastatic sites like CRLM, type I collagen is the most prevalent, forming robust fibrillar structures that confer tensile strength to tissues. It is particularly abundant in the liver’s Glisson capsule, the blood vessel walls, and perisinusoidal space (Space of Disse), where it supports hepatocytes and regulates cell migration [82,83]. Type III collagen, often co-localized with Type I, is crucial for maintaining liver tissue elasticity and architecture. In the setting of metastasis, increased deposition of Types I and III collagen is observed, leading to a stiffened matrix that promotes cancer cell invasion and angiogenesis [84]. Another important form, Type IV collagen, forms part of the basement membrane of hepatocytes and endothelial cells, a specialized ECM that separates epithelial cells from the underlying connective tissue. In both liver and metastatic colorectal liver cancer, Type IV collagen plays a role in maintaining the integrity of sinusoidal endothelium and basement membranes, influencing cancer cell extravasation and colonization [85]. Additionally, Type VI collagen, often upregulated in liver fibrosis and cancer, contributes to ECM remodeling, enhancing metastatic niche formation through the recruitment of stromal cells and promotion of angiogenesis [86] [Table 2].

Elastin, a highly stable ECM liver protein, is critical in organs that require elasticity, such as hepatic arteries, also contributing to vascular integrity [87,88]. Though its turnover is generally low, elastin may be remodeled in pathological conditions, such as hepatic fibrosis, contributing to disease persistence due to its stability [89]. Laminins are crucial for hepatocyte polarity and attachment to the sinusoidal endothelial basement membrane [90].

Hyaluronan (HA), also known as hyaluronic acid, is a critical component of the liver ECM, being the most abundant GAG, predominantly synthesized by hepatic stellate cells. Its main role is the regulation of tissue hydration and inflammation [91]. During CRLM, the accumulation of HA is associated with ECM remodeling, providing a matrix that enhances cancer cell motility and immune evasion. High levels of HA in the tumor stroma have been correlated with poor prognosis in liver metastasis patients [92,93]. As expressed by Lopes et al. in 2020 [94], HS is abundant in the basement membranes of blood vessels, playing a role in liver regeneration and tumor progression. In CRLM, heparan sulfate proteoglycans, such as syndecan-1 and glypican-3, facilitate cancer cell proliferation and metastasis by sequestering growth factors like hepatocyte growth factor (HGF) and fibroblast growth factor (FGF), which activate pro-tumorigenic pathways. Targeting HS-modulated pathways has become a focus of therapeutic strategies aiming to disrupt cancer progression in the liver [93]. Proteoglycans play key roles in tissue architecture and growth factor signaling. One of the most important in liver and CRLM is decorin, a small leucine-rich proteoglycan that binds to collagen fibers and regulates ECM assembly. In CRLM, loss of decorin expression has been linked to enhanced tumor invasion and ECM disorganization, leading to a more permissive microenvironment for cancer progression.

### 4.2. ECM Remodeling in Normal and Pathological Liver Conditions

In a healthy liver, ECM remodeling is a tightly regulated process essential for maintaining tissue homeostasis and facilitating liver regeneration. HSCs, Kupffer cells (KCs), and endothelial cells play pivotal roles in ECM production and degradation. Under normal conditions, ECM turnover ensures a balanced deposition and removal of ECM components, enabling the liver to adapt to metabolic demands and repair minor injuries [95,96,97].

However, pathological conditions such as chronic liver fibrosis and cirrhosis disrupt this balance, leading to excessive ECM deposition [95]. Activated HSCs become the primary source of ECM overproduction, particularly collagen types I and III. This fibrogenic response results in the stiffening of the liver parenchyma, impairing liver function and promoting further disease progression. Additionally, ECM degradation is also dysregulated, with MMPs and TIMPs playing key roles in this process [98,99]. During hepatocellular carcinoma (HCC) development, ECM remodeling is further exacerbated [100]. Tumor cells and the tumor microenvironment contribute to aberrant ECM dynamics, facilitating tumor growth, angiogenesis, and metastasis. The interplay between cancer cells and the altered ECM creates a permissive environment for tumor progression and dissemination [101].

In CRLM development, the ECM is fundamental. Initially, via ECM-elastin remodeling, it facilitates the seeding of metastatic CRC cells in the hepatic microenvironment, providing anchorage and biochemical support. This remodeling impairs normal liver function and creates a pro-tumorigenic niche, allowing cancer cells to thrive. As the metastatic niche evolves, the ECM not only sustains tumor growth but also actively participates in modulating the local immune response, angiogenesis, and stromal cell recruitment. Fibronectin has been demonstrated to be crucial. Fibronectin exists in both soluble and insoluble forms. Soluble fibronectin is produced by hepatocytes and circulates in the plasma, while insoluble fibronectin is assembled into fibrils within the ECM. In metastatic liver disease, fibronectin interacts with integrins on the surface of cancer cells, promoting their adhesion to the liver’s ECM and supporting angiogenesis and cancer cell survival [100]. Overexpression of fibronectin in CRLM has been linked to enhanced tumor aggressiveness and chemoresistance, as it acts as a scaffold for both cancer cells and tumor-associated fibroblasts (TAFs) [102]. In CRLM, cancer cells often exploit laminin-binding integrins (e.g., α6β1 and α6β4) [103] to facilitate their invasion and anchorage in the liver. The laminin-rich ECM provides not only a structural base but also signaling cues that support tumor survival and metastasis. Laminin isoforms, such as Laminin-511 and Laminin-332 [104], are known to influence tumor invasion by enhancing the migratory potential of colorectal cancer cells in the liver [Table 3]. Moreover, laminins engage with diverse cell populations in the tumor microenvironment, such as endothelial and immune cells. Notably, LAMA5 expression can be enhanced in response to inflammatory signals from tumor-associated myeloid cells through the TNFα/NFκB signaling pathway. This upregulation contributes to angiogenesis and influences Notch signaling in endothelial cells, thereby promoting tumor progression [105,106].

### 4.3. Influence of ECM Stiffness and Composition on Cellular Behavior

Mechanotransduction refers to the process by which cells sense and respond to mechanical cues from the ECM [107]. Liver ECM stiffness, driven by collagen cross-linking and deposition, profoundly affects cellular behavior, influencing proliferation, differentiation, migration, and apoptosis [108,109]. In a healthy liver, ECM elasticity supports hepatocyte function, whereas increased stiffness in fibrosis, cirrhosis, and cancer activates mechanosensitive pathways, altering cellular responses. Cells detect ECM stiffness via integrin receptors, which relay mechanical signals to the cytoskeleton, modulating gene expression. In liver fibrosis, ECM stiffening induces YAP/TAZ nuclear translocation, promoting genes linked to proliferation, migration, and survival [110]. This response is mediated by the RhoA/ROCK pathway, which regulates cytoskeletal tension and FAK activation [106]. YAP/TAZ also drives profibrotic gene expression, sustaining HSC activation and fibrosis progression [111]. Furthermore, these transcription factors contribute to cancer progression, supporting tumor cell survival and invasion in stiffened microenvironments like CRLM [112,113]. Beyond mechanical cues, biochemical interactions regulate cell behavior. ECM components such as laminins and fibronectins engage integrins, activating ERK/MAPK and PI3K/AKT pathways to promote hepatocyte survival [114,115]. Proteoglycans modulate growth factor availability, facilitating angiogenesis, tissue remodeling, and tumor proliferation, reinforcing the role of ECM in fibrosis and metastasis [116].

### 4.4. Interaction Between CRC Cells and Hepatic ECM

The reciprocal involvement of CRC cells and the hepatic ECM affects numerous cellular processes, including adhesion, migration, and pre-metastatic niche (PMN) formation. Additionally, as already described above, liver ECM is rich in cytokines, including IL-6 and TGF-β, which are crucial in regulating the behavior of CRC cells. Indeed, IL-6 activates signaling pathways that induce EMT, thereby enhancing the invasive potential of CRC cells [117]. TGF-β, on the other hand, drives the transformation of HSCs and promotes ECM remodeling, further facilitating metastatic progression [118]. CRC cells initiate metastatic colonization by adhering to hepatic ECM proteins, primarily through integrins such as α1β1, α2β1, α6β1, and α3β1. Integrin α1β1 primarily binds collagen types I, III, IV, IX, and XVI, exhibiting a preference for non-fibrillar collagen, especially collagen IV. However, it can also engage with fibrillar collagens through specific sequences, such as GFOGER, present within the collagen triple helix [119]. Integrin α2β1, also referred to as VLA-2, preferentially binds to fibrillar collagens, including types I, III, and V, while also recognizing other collagen types such as XIV and XVI. Its interaction with collagen is facilitated by specific sequences that enable effective binding [114]. Integrin α6β1 has a high binding affinity for laminin isoforms, in particular laminin-511 and laminin-332. This interplay engages multiple binding sites, including the γ1 chain of laminin, which engages the metal ion-dependent adhesion site (MIDAS) on the β1 subunit of integrin. This binding is required for epithelial cell adhesion to the basement membrane and influences signaling pathways that promote cell survival and migration [120]. Likewise, integrin α3β1 interacts with laminins, specifically recognizing their C-terminal domains. It is involved in several cellular functions, such as adhesion and migration, which are facilitated by its engagement with laminins, which are fundamental components of the ECM. Collectively, these integrins contribute to the dynamical regulation of ECM–cell interactions, influencing processes essential for tissue integrity and repair [121]. Additionally, the ECM dynamic remodeling during metastatic progression, facilitated by MMPs, allows CRC cells to penetrate deeper into the hepatic parenchyma, establishing secondary tumors. In more detail, MMPs facilitate hepatic ECM, targeting collagen(s), fibronectin, and laminins. This degradation disrupts the ECM’s structural framework, thereby enabling CRC invasion. Specifically, MMP-2, MMP-7, and MMP-9, which are often overexpressed in CRC cells, contribute to the breakdown of type IV collagen, a vital element of the basement membrane [122] [Table 4]. These pathways also generate bioactive fragments that further promote CRC cell survival and invasion [123].

CRC cells could also actively engage with hepatic HSCs and KCs, both of which play crucial roles in remodeling the liver’s ECM via specific cytokines, chemokines, and growth factor release. This creates a pro-tumorigenic microenvironment. The presence of CRC cells drives the activation and transition of HSCs from a quiescent state to a myofibroblast-like phenotype, known as activated-HSCs (aHSCs) [124]. Factors such as transforming growth factor-beta (TGF-β), which can be secreted by CRC cells or other elements of the tumor microenvironment, are the main inducers of this activation [125]. Once activated, HSCs remodel the hepatic ECM by synthesizing collagen and other matrix components, leading to increased stiffness and changes in matrix composition through the CXCR4/TGF-β1 axis [126].

## 5. ECM Remodeling the Metastatic Niche

A considerable body of research has been published on the influence of soluble factors derived from CRC on the liver microenvironment, which contributes to pre-metastatic niche (PMN) formation through ECM remodeling [Figure 2]. Specific studies on CRLM have shown that chemokine-ligand 2 (CCL2) mobilizes immature myeloid cells to the liver, where they produce matrix metalloproteinases (mainly MMP-2 and MMP-9), facilitating ECM degradation and enhancing metastatic spread [127]. Recent evidence suggests that CCL2, encapsulated in EVs, induces the differentiation of macrophages into an M2 pro-tumorigenic phenotype while simultaneously promoting liver fibrosis, thus making the ECM more susceptible to cancer cell invasion [128]. Additionally, EV-packaged TIMP1 has been reported to initiate a positive feedback loop by upregulating TIMP1 expression in liver fibroblasts, resulting in ECM remodeling, as demonstrated in CRC models such as HCT116, HT29, and SW620 [129]. Furthermore, EV-mediated transfer of miR-122-5p has been shown to regulate PMN formation by modulating cell–ECM adhesion; it upregulates N-cadherin and vimentin while downregulating E-cadherin in hepatocytes, as seen in non-small cell lung cancer (NSCLC) models [130].

There have also been significant insights into the role of gastric cancer-derived EVs, particularly EV-packaged EGFR, in modifying liver stromal cells. These EVs translocate EGFR to the plasma membrane of hepatic stromal cells, subsequently activating HGF/c-Met signaling pathways, thereby supporting cancer cell invasion and liver colonization [131]. Collectively, these studies underscore the profound impact of primary tumor-derived soluble factors and EVs on remodeling the liver’s ECM and immune stroma to support metastasis. Targeting EV-mediated signaling and the fibrotic niche represents a promising therapeutic avenue for early intervention in cancers prone to liver metastasis.

CAFs play a pivotal role in liver metastasis by extensively remodeling the ECM [132]. They deposit structural proteins like collagen and fibronectin while degrading existing components via MMPs, leading to a denser, stiffer ECM that facilitates tumor invasion. CAFs also induce EMT, enhancing cancer cell migration [132]. Colinge et al. [133] identified two key CAF subtypes in CRLM: contractile CAFs and ECM-remodeling/pro-angiogenic CAFs, the latter linked to stiff ECM regions and low-proliferating tumor cells. Targeting ECM-remodeling CAFs using LTBP2-specific antibodies, which regulate collagen biosynthesis, presents a promising therapeutic avenue, particularly for CRLM and intrahepatic cholangiocarcinomas. Lai et al. [134] further detailed CAF heterogeneity, categorizing subtypes (iCAF, myCAF, ECM-CAF, Ctr-CAF, Cluster 3-CAFs) and their dual pro- and anti-tumorigenic roles. Beyond ECM modification, CAFs secrete growth factors like TGF-β and HGF, driving invasion, angiogenesis, and immune evasion. They also reorganize the ECM to form migration pathways while promoting desmoplastic reactions that inhibit immune infiltration. Developing advanced 3D models is crucial to studying CAF heterogeneity and refining targeted therapies, addressing their role in metastasis and tumor progression.

## 6. ECM and Chemoresistance

The ECM, once considered a passive scaffold, is now recognized for actively promoting tumor resilience and chemoresistance through physical obstruction, pro-survival signaling, and drug sequestration. Understanding these mechanisms is vital for improving therapeutic strategies [49,135,136]. Tumoral ECM is profoundly different from that ‘in normal tissues, both in composition and mechanical properties, being typically denser and stiffer. Breast cancer ECM can exhibit stiffness up to ten times greater than normal breast tissue, significantly increasing interstitial fluid pressure (IFP) [137]. This elevated IFP impedes the effective distribution of therapeutic agents within the tumor. Moreover, CAFs and other cells within the tumor microenvironment (TME) drive quantitative and functional alterations in ECM components, further exacerbating these challenges [138]. The ECM serves as a physical barrier, shielding tumor cells from therapeutic agents by creating a dense, rigid network that restricts drug infiltration. This encapsulation often leads to hypoxia and metabolic stress, which activate pathways associated with drug resistance [126]. Additionally, the ECM’s structure imposes diffusion limitations, particularly for nanoparticles and macromolecular drugs, hindering their ability to penetrate deeply into tumor tissues [138,139]. Additionally, the ECM provides structural and biochemical support for cancer stem cells (CSCs), promoting their survival, self-renewal, and chemotherapy resistance [140]. ECM-CSC interactions remodel the ECM and activate survival pathways like integrin and PI3K, enhancing invasiveness and apoptosis resistance. In particular, increased cell–ECM interactions activate several survival-signaling pathways, including integrin signaling, Rho-ROCK, and PI3K pathways. These pathways promote cellular processes such as EMT [141], which enhances the invasive potential of cancer cells and their resistance to apoptosis.

CRLM ECM undergoes extensive remodeling, which is critical in inducing chemoresistance. The fibrotic and collagen-rich ECM, characteristic of CRLM, significantly increases tissue stiffness and interstitial pressure [142], forming a dense physical barrier that impedes the diffusion of chemotherapeutic agents into the tumor core [143,144,145]. For instance, proteomic analysis revealed that 19 out of 22 collagen-α chains are significantly upregulated in CRLM tissues compared to healthy liver tissues, indicating increased collagen production in metastatic cells. Additionally, second-harmonic generation (SHG) imaging has been utilized to assess collagen I characteristics in rectal cancer, highlighting the potential of SHG in evaluating collagen alterations in cancerous tissue [146]. The abnormal vasculature associated with CRLM further exacerbates these issues. Tumor vessels are irregular, leaky, and poorly perfused, leading to heterogeneous drug delivery, as evidenced by intravital imaging studies that demonstrated that the abnormal vasculature in tumors results in an uneven distribution of chemotherapeutic agents, with hypoxic regions receiving lower drug concentrations than well-perfused areas. This uneven distribution can reduce the efficacy of chemotherapy in hypoxic tumor regions [147]. These physical barriers contribute to suboptimal drug concentrations, allowing chemoresistant subclones to thrive.

Beyond mechanical hindrances, the hepatic ECM modulates biochemical interactions that directly promote cancer cell survival under chemotherapeutic stress. ECM components such as fibronectin, laminin, and collagen engage integrin receptors in cancer cells, activating survival pathways [148]. Integrin-mediated activation of the FAK pathway has been particularly implicated in chemotherapy resistance. These interactions can induce the expression of anti-apoptotic proteins and inhibit pro-apoptotic signals, helping cancer cells withstand the cytotoxic effects of chemotherapy. For instance, the binding of integrins to ECM components can activate the FAK pathway, which is known to enhance cell survival and resistance to various stressors, including chemotherapy [149]. Specific signaling pathways involved in ECM-induced chemoresistance include the PI3K/AKT and ERK/MAPK pathways. Upon ECM–integrin binding, PI3K is activated, leading to the production of PIP3 and subsequent activation of AKT. AKT promotes cell survival and growth by phosphorylating and inhibiting pro-apoptotic factors like BAD and caspase-9 while activating mTOR, which drives protein synthesis and cell proliferation. The PI3K/AKT pathway also enhances metabolic adaptation and drug efflux, further contributing to chemoresistance [149].

Similarly, the ERK/MAPK pathway is activated through integrin-mediated recruitment of receptor tyrosine kinases, leading to the activation of the RAS-RAF-MEK-ERK signaling cascade. ERK translocates to the nucleus and promotes the expression of genes involved in cell proliferation, survival, and differentiation. This pathway enhances the expression of anti-apoptotic proteins, such as BCL-2 and MCL-1, and inhibits pro-apoptotic signals, providing a survival advantage to cancer cells in the face of chemotherapeutic stress [150]. Additionally, the PI3K/AKT axis stimulates mTOR activity, enhancing protein synthesis and metabolic reprogramming to meet the energy demands of resistant cancer cells.

The ERK/MAPK (extracellular signal-regulated kinase/mitogen-activated protein kinase) pathway is another critical mediator of ECM-induced chemoresistance. The activation of integrins and other ECM receptors leads to the recruitment and activation of receptor tyrosine kinases (RTKs), which subsequently activate the RAS-RAF-MEK-ERK signaling cascade. ERK translocates to the nucleus and promotes the expression of genes involved in cell proliferation, survival, and differentiation. This pathway enhances the expression of anti-apoptotic proteins, such as BCL-2 and MCL-1, and inhibits pro-apoptotic signals, providing a survival advantage to cancer cells in the face of chemotherapeutic stress [151,152].

Furthermore, TGF-β signaling, modulated by ECM stiffness and composition, drives EMT, which enhances cell motility and drug resistance by upregulating survival proteins like vimentin and N-cadherin and suppressing epithelial markers like E-cadherin. In CRLM mouse models, TGF-β inhibition restored chemosensitivity, reducing tumor burden when combined with standard chemotherapy [153,154].

Integrating CRISPR screening with metabolic analyses can reveal regulators, such as those modulating the PI3K/AKT/HIF1A pathway, that contribute to drug resistance [155]. Using in vivo CRISPR/Cas9 screening to identify CHSY1 as a key driver of CRLM has revealed that CHSY1 overexpression leads to CD8+ T cell exhaustion by activating the succinate metabolism pathway, which in turn triggers the PI3K/AKT/HIF1A cascade and upregulates PD-L1 expression. Notably, CHSY1 knockdown, and its pharmacological inhibition by artemisinin, not only diminishes metastatic progression but also synergizes with anti-PD1 therapy [156].

CRISPR-Cas9 has also been used in mouse intestinal tumor organoids derived from Apc and Kras mutant mice, which were then transduced with a pooled gRNA library targeting several putative tumor suppressors to functionally validate candidate CRC driver genes. Following orthotopic transplantation into immunodeficient mice, the selective overrepresentation of specific gRNAs in emerging tumors signaled the loss of the corresponding gene’s function, thereby implicating these genes in tumor suppression [157]. These comprehensive workflows underscore how CRISPR-based approaches could uncover novel mediators of ECM-driven chemoresistance and inform combinatorial strategies to enhance therapeutic efficacy in CRLM.

## 7. Methods for Studying ECM in Cancer Research

Decellularization techniques, originally developed for transplantation purposes, have been now widely applied in oncology to create ECM scaffolds [158]. Decellularization involves the removal of cellular components from tissues while preserving the ECM’s native structure and composition. To effectively decellularize tissue, various methods have been employed, such as detergents, enzymatic digestion, or a combination of both, always ensuring the removal of cellular components while preserving the structural integrity and composition of the “original” ECM [159]. The decellularized liver scaffolds maintain critical ECM elements, including collagen, laminin, and fibronectin, which are essential for recreating the liver-specific microenvironment. Mazza et al., in 2015, demonstrated the complete decellularization of the whole human liver to form an ECM scaffold with preserved architecture, retaining key ECM proteins such as collagen, laminin, and fibronectin [160]. Additionally, research on decellularized liver matrices (DLMs) has shown the retention of ECM pivotal proteins for liver structure and physiology (including collagen IV, fibronectin, and laminin) [161]. Moreover, decellularized ECM from the peritoneal cavity has been utilized to support the growth of organoids derived from peritoneal metastases, highlighting the versatility of decellularization techniques in creating tumor-specific microenvironments [162].

By mimicking the stiffness, composition, and architecture of the native hepatic ECM, these scaffolds provide a physiologically relevant environment that enables the investigation of cancer cell invasion, proliferation, and drug response. Their ability to replicate the tumor microenvironment allows for more accurate modeling of liver metastasis and facilitates the identification of potential therapeutic targets, thus offering a more accurate representation of in vivo conditions [138]. Acellular liver scaffolds have provided insights into how CRC cells remodel the ECM by upregulating the deposition of collagen I, fibronectin, and other matrix proteins. This remodeling leads to increased ECM stiffness, which not only promotes CRC cell adhesion and invasion but also activates integrin-mediated signaling pathways like PI3K/AKT and FAK. These pathways contribute to enhanced cell survival by inhibiting apoptotic processes and facilitating tumor cell proliferation. Furthermore, the remodeled ECM in these models creates a pro-metastatic niche that supports CRC cell evasion of chemotherapy, notably by upregulating the expression of drug-resistant proteins such as P-glycoprotein and MRP1. This creates a dynamic, tumor-supportive microenvironment that limits the efficacy of chemotherapeutic agents and fosters chemoresistance [139].

In contrast to two-dimensional (2D) cultures, where cells are exposed to planar and artificial surfaces, three-dimensional (3D) liver scaffolds provide a biologically relevant matrix that delivers both mechanical and biochemical signals, thereby modulating cellular behavior and drug responsiveness. Bioprinted constructs integrating liver ECM and CRC cells emulate the spatial organization of the hepatic microenvironment, as demonstrated in prior studies [140]. The 3D architecture of decellularized liver ECM scaffolds enhances the complexity of cellular interactions, supports cell viability, and promotes invasive capabilities when compared to conventional 2D culture systems. This advanced structure facilitates the activation of integrin-mediated signaling pathways, including the FAK/PI3K/AKT and RAS-RAF-MEK-ERK cascades, further underscoring its relevance in mimicking physiological and pathological liver conditions.

Another major advantage of the 3D scaffold system is its capacity to replicate the increased mechanical stiffness of the liver observed in CRLM, a defining feature of fibrosis and metastatic progression. This stiffness contributes to altered cellular signaling and metabolic reprogramming, facilitating the activation of mechanotransduction pathways that are minimally engaged in 2D cultures. CRC cells cultured within these 3D liver scaffolds demonstrate markedly elevated expression of MMPs, particularly MMP-2 and MMP-9, which are pivotal in the degradation of collagen and other ECM components. This ECM remodeling process is essential for CRC cell invasion and metastasis, and the use of 3D scaffolds provides a more accurate model for studying ECM turnover within the context of metastatic progression. Furthermore, unlike 2D cultures, where drug penetration is uniform, in 3D scaffolds, CRC cells are exposed to more physiologically relevant gradients of drugs, oxygen, and nutrients, which better replicates the conditions in solid tumors. This enables the study of differential drug resistance in areas of low perfusion or high ECM stiffness, providing insights into how tumors evade chemotherapy and enabling more precise testing of therapeutic agents [163,164,165,166].

## 8. Therapeutic Implications and Future Directions

Targeting the ECM directly addresses its role in CRLM by disrupting pro-tumorigenic remodeling and improving drug delivery [167,168,169]. MMPs play a significant role in ECM degradation, facilitating cancer cell invasion. Early MMP inhibitors reduced metastatic spread in animal models but failed in clinical trials due to off-target toxicity and low specificity [170]. More recently, selective inhibitors targeting MMP-9 have shown efficacy in reducing metastatic colonization in preclinical CRC models. LOX, responsible for collagen cross-linking and ECM stiffening, has been identified as a key driver of tumor invasiveness. Studies using beta-aminopropionitrile (BAPN) demonstrated reduced ECM stiffness and improved drug delivery in CRC metastases to the liver [92]. LOX inhibition also impairs pre-metastatic niche formation, highlighting its potential for early intervention.

Table 5 aims to bridge the translational gap between preclinical evidence and clinical application, systematically summarizing ongoing trials from ClinicalTrials.gov, and providing a clear snapshot of how these approaches are being actively pursued in the clinical setting.

Biomarkers indicative of ECM remodeling, such as elevated MMP-9 and LOX levels, are strongly correlated with poor outcomes in CRC. MMP-9 facilitates cancer cell invasion and metastatic progression. Increased serum and tissue levels of MMP-9 have been associated with enhanced tumor aggressiveness and reduced patient survival rates [171,172]. Similarly, LOX promotes a pro-tumorigenic microenvironment. High LOX expression has been linked to greater matrix rigidity, heightened metastatic potential, and adverse clinical outcomes in CRC patients.

Furthermore, integrin α5β1 has emerged as a significant biomarker, particularly in the context of chemoresistance. Integrin α5β1 is known to mediate cell–ECM adhesion, survival signaling, and resistance to therapy. Overexpression of integrin α5β1 in chemoresistant CRC tumors highlights its predictive value for therapeutic response. Some studies have demonstrated that integrin α5β1 promotes cancer cell survival by activating downstream signaling pathways, such as FAK and PI3K/AKT, thereby supporting tumor progression and evasion of apoptosis [173,174]. This integrin is also implicated in maintaining cancer stem cell properties, further contributing to resistance to conventional chemotherapies [175,176]. Additionally, integrin α5β1’s role in ECM remodeling and interaction with fibronectin reinforces its potential as a prognostic biomarker for CRC progression [170,171,172,173,174,175,176,177].

Combining LOX inhibitors with chemotherapy represents a promising strategy to overcome ECM-mediated barriers in CRLM. Preclinical studies have demonstrated that LOX inhibition using agents such as BAPN or specific LOX-targeting drugs significantly reduces ECM stiffness. This mechanical modulation enhances vascular permeability and facilitates deeper penetration of chemotherapeutic agents into metastatic lesions, thereby improving therapeutic efficacy [176]. Similarly, the co-administration of integrin antagonists with chemotherapy disrupts cell–ECM adhesion mechanisms mediated by integrin αvβ3 and α5β1 interactions with fibronectin and vitronectin. This disruption not only impairs tumor cell anchorage but also attenuates pro-survival signaling pathways, including FAK and PI3K/AKT. As a result, tumors become more susceptible to cytotoxic agents. Notably, integrin inhibition has been shown to synergize with chemotherapeutics, increasing apoptosis and reducing tumor burden in preclinical CRLM models [178,179,180].

Nanoparticles have emerged as powerful tools for targeted drug delivery in metastatic colorectal cancer, offering precise localization and controlled release of therapeutic agents in metastatic niches. Their small size and customizable surface properties allow nanoparticles to penetrate the dense ECM and target tumor-specific microenvironments with high specificity. For instance, functionalizing nanoparticles with ECM-binding peptides, such as RGD sequences, facilitates binding to integrins highly expressed in metastatic colorectal liver lesions, enhancing drug accumulation within the tumor site. Additionally, nanoparticles can be engineered to respond to the unique biochemical and mechanical properties of the tumor microenvironment, such as pH, enzymatic activity, or redox conditions, enabling on-demand drug release [181].

The future of ECM scaffolds in preclinical testing is poised for transformative advancements, driven by rapid innovations in biomaterials and fabrication technologies. The integration of ECM scaffolds with cutting-edge platforms, such as 3D bioprinting and organ-on-a-chip systems, promises to create even more sophisticated models that faithfully capture the dynamic complexities of the tumor microenvironment.

Yet, several critical hurdles remain, such as achieving standardization in ECM scaffold production (which is imperative to ensure consistency, reproducibility, and broader adoption across both academic research and industrial applications) or incorporating the full complexity of the ECM, including its biochemical diversity and mechanical properties.

These challenges represent opportunities to redefine the frontiers of cancer research and precision medicine. By overcoming these barriers, ECM scaffolds could revolutionize the way we study tumor biology and, in particular, CRLM, opening unprecedented avenues for developing personalized therapeutic strategies and predictive testing models.

## 9. Conclusions

In conclusion, our study accurately presents the complex role of the liver ECM CRLM. We have shown that ECM remodeling, driven by factors such as MMPs and LOX, not only facilitates tumor invasion and chemoresistance but also creates a dynamic microenvironment that actively shapes cancer progression. These findings are significant because they extend and refine current theories on metastatic dissemination while challenging the limitations of traditional therapeutic approaches. Our work underscores that targeting the ECM, whether through selective inhibitors or novel nanoparticle-based drug delivery systems, holds real-world promise for improving patient outcomes. This research offers practical insights for clinicians by identifying biomarkers like MMP-9, LOX, and integrin α5β1 that could inform both therapeutic strategies and clinical decision-making. Moreover, by clearly delineating unresolved questions, such as the optimal integration of ECM-targeted therapies with existing treatments, we invite future investigations to build upon these results, ultimately advancing personalized treatment strategies in oncology. As shown by the reported ongoing studies, nanoparticle-based drug delivery and advanced 3D ECM scaffold models are poised to further revolutionize our understanding of CRLM and open unprecedented avenues for personalized therapeutic strategies and predictive testing models. Thus, our conclusion not only encapsulates the critical findings of our research but also sets a new direction for exploring ECM-mediated pathways in cancer, reaffirming the importance of innovative approaches in transforming both scientific understanding and clinical practice.

## Figures and Tables

**Figure 1 cancers-17-00953-f001:**
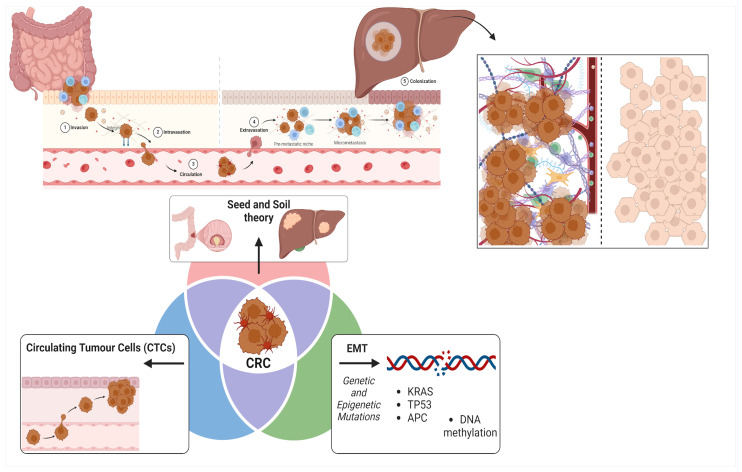
Colorectal liver metastatic cascade.

**Figure 2 cancers-17-00953-f002:**
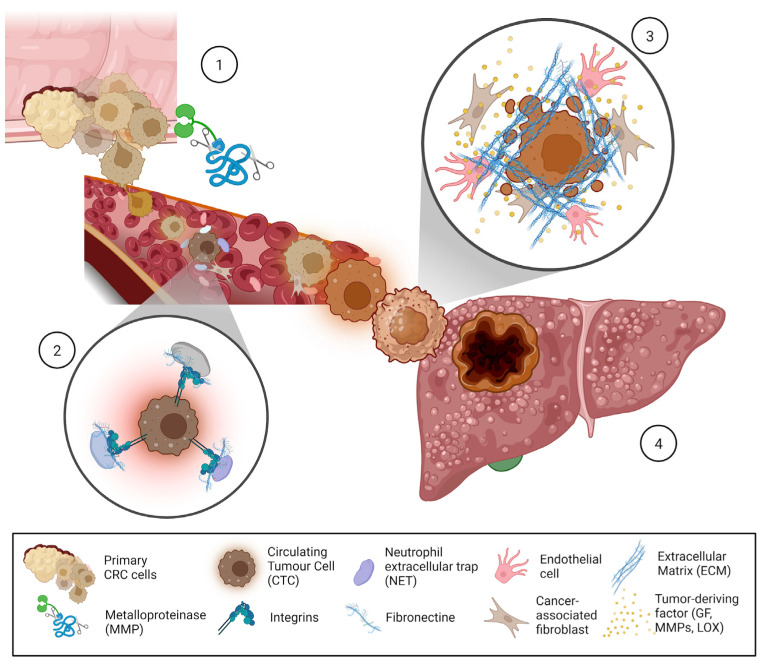
Pre-metastatic Liver Niche Formation. (1) Angiogenesis and intravasation: High angiogenic and matrix metalloproteinase (MMP) activity at the primary tumor site leads to vascular disruption, facilitating tumor cell intravasation and entry into circulation. Circulating tumor cells (CTCs) may secrete extracellular matrix (ECM) components to evade immune surveillance. (2) Interaction with neutrophils and NETs: CTCs interact with neutrophil extracellular traps (NETs) and NETotic neutrophils via integrin-mediated matrix-like connections, enhancing their survival in circulation. (3) Endothelial cell remodeling and CTC attachment: Endothelial cells (ECs) assemble fibrillar fibronectin, promoting CTC adhesion to the endothelial wall at distant sites via CAF action. Elevated MMP activity disrupts vascular integrity, facilitating CTC extravasation. Tumor-derived factors, including growth factors, MMPs, lysyloxidase (LOX), and ECM proteins (e.g., fibronectin), create a liver pre-metastatic niche. (4) Dormancy and metastasis initiation: CTCs that extravasate into the liver may enter dormancy. Proteases within NETs, such as neutrophil elastase and MMP-9, cleave laminin to generate specific matrikines that can awaken dormant tumor cells. These ECM remodeling processes collectively support the establishment and progression of metastasis.

**Table 1 cancers-17-00953-t001:** Key ECM proteins and molecules in healthy liver.

Molecule/Protein Name	Molecular Structure (Subtypes)	Biological Function in Healthy Liver	Tissue Distribution in Liver
Collagen Type I	Fibrillar (I, III)	Provides tensile strength, structural integrity	Perisinusoidal space
Collagen Type IV	Non-fibrillar	Maintains basement membrane integrity, supports cell adhesion	Sinusoidal endothelium
Elastin	Fibers	Provides tissue elasticity and maintains vascular integrity	Blood vessels
Fibronectin	Glycoprotein	Facilitates cell adhesion, migration, and tissue repair	ECM, hepatocytes
Laminin	Heterotrimer (α, β, γ)	Supports cell adhesion and differentiation	Basement membrane of sinusoids
Hyaluronic acid	Non-sulfated GAG	Regulates hydration and cellular motility	Hepatic stellate cells
Heparan sulfate	Sulfated GAG	Modulates growth factor signaling and maintains liver homeostasis	Basement membrane
Decorin	Proteoglycan	Controls collagen fibrillogenesis	ECM

**Table 2 cancers-17-00953-t002:** Key ECM proteins and molecules in colorectal liver metastasis (CRLM).

Molecule/Protein Name	Molecular Structure (Subtypes)	Biological Function in CRLM	Tissue Distribution in Metastasis
Collagen Type I	Fibrillar (I, III)	Increases rigidity, promotes cancer invasion	Tumor stroma
Collagen Type IV	Non-fibrillar	Facilitates cancer cell extravasation and colonization	Tumor basement membrane
Elastin	Fibers	Increased vascular stiffness, promotes abnormal vasculature	Metastatic blood vessels
Fibronectin	Glycoprotein	Enhances tumor adhesion, migration, and angiogenesis	Tumor ECM
Laminin	Heterotrimer (α, β, γ)	Facilitates tumor invasion and survival in metastatic sites	Metastatic niche
Hyaluronic acid	Non-sulfated GAG	Promotes cell motility, immune evasion	Tumor stroma
Heparan sulfate	Sulfated GAG	Sequester growth factors, enhance tumor growth	Tumor-associated basement membrane
Decorin	Proteoglycan	Loss of decorin leads to enhanced tumor invasion	Tumor ECM

**Table 3 cancers-17-00953-t003:** Key laminins in healthy liver and colorectal liver metastasis (CRLM).

Laminin Type/Isoform	Chains (α, β, γ)	Role in Healthy Liver	Role in CRLM
Laminin-111	α1, β1, γ1	Present during early liver development; involved in hepatocyte polarization and basement membrane formation	Rare in adult liver, but cancer cells may exploit it for metastatic niche establishment
Laminin-211	α2, β1, γ1	Important for liver regeneration and muscle maintenance	Rarely found in CRLM; minimal role in metastasis
Laminin-332 (formerly 5)	α3, β3, γ2	Not highly expressed in normal liver tissue	Promotes cancer cell migration, invasion, and metastasis through integrin signaling (α6β4 integrin)
Laminin-411	α4, β1, γ1	Important for liver vasculature integrity; contributes to sinusoidal endothelium and ECM maintenance	Involved in vascular remodeling and angiogenesis in CRLM, providing a supportive environment for tumor growth
Laminin-511	α5, β1, γ1	Supports hepatocyte attachment to ECM and maintains normal liver structure	Upregulated in the metastatic liver microenvironment, enhances the adhesion and migration of colorectal cancer cells
Laminin-521	α5, β2, γ1	Maintains structural integrity of the liver’s vascular basement membrane	Involved in metastatic cancer cell colonization, promoting survival of colorectal cancer cells in liver metastases
Laminin-332	α3, β3, γ2	Rare in normal adult liver, mostly in basal membranes	Facilitates cancer cell invasion and metastasis through interaction with integrins (e.g., α6β4) and promotes tumor cell survival in the metastatic niche
Laminin-111 (Developmental)	α1, β1, γ1	Expressed during liver development; aids in tissue differentiation and basement membrane formation	Abnormally re-expressed in some liver metastases, contributing to the establishment of a supportive metastatic niche

**Table 4 cancers-17-00953-t004:** MMP expression in healthy liver and CRLM.

MMP	Healthy Liver	Colorectal Liver Metastasis (CRLM)
MMP-2	Present in latent form; involved in normal ECM turnover	Overexpressed and activated; facilitates tumor cell invasion leading to the breakdown of ECM barriers
MMP-7	Low expression; limited role in normal liver function	Upregulated; associated with aggressive tumor behavior, progression, and poor prognosis
MMP-9	Detected in latent form; participates in ECM maintenance	Elevated levels, especially in active form; promote metastasis

**Table 5 cancers-17-00953-t005:** Summary of therapeutic studies targeting ECM in CRLM (resource: ClinicalTrials.gov).

Trial ID/ Study References	Target	Drug/ Intervention	Type	N *	Key Findings	Status/ Results
NCT04755907	3D colorectal cancer models and organoids	Same chemotherapy drugs as the corresponding patients	Preclinical	120	Response to adjuvant chemotherapy evaluated according to DFS	unknown
NCT03131778	MMP9	Laparoscopic vs open liver resection	Clinical	40	Evaluate differences in inflammatory response	completed
NCT00835679	EGFR FAK	Cetuximab Dasatinib	Clinical	9	Reduction in at least 1 biomarker of the pathway inhibited	closed prematurely (slow accrual)
NCT01008475	integrin αν heterodimers	EMD 525797 (Abituzumab) with cetuximab and Irinotecan	Clinical	232	Assess the tolerability of different doses (phase I) and explore the efficacy and tolerability (phase II)	completed, primary PFS endpoint not met, accepted tolerability
NCT03170960	tyrosin kinase (PD-1)	Cabozantinib (XL184)	Clinical	1732	Assess safety, tolerability, preliminary efficacy, and pharmacokinetics	active, not recruiting
NCT02837263	tyrosin kinase (PD-1)	Pembrolizumab with stereotactic body radiotherapy (SBRT)	Clinical	18	Recurrence rate at 1 year following clearance of metastatic disease	completed
NCT04508140	tyrosin kinase (PD-1)	BO-112 with Pembrolizumab	Clinical	18	Reverse the primary resistance that microsatellite stability presents to the PD-1 inhibitors	terminated, low recruitment rate
NCT04046445	tyrosin kinase (PD-1) and IgG4	TP128 and BI 754091 (Ezabenlimab)	Clinical	96	Evaluate the safety and tolerability	active, not recruiting
NCT02298946	tyrosin kinase (PD-1)	AMP-224 plus SBRT	Clinical	17	Evaluate whether the anti-tumor immunity can be enhanced by radiation therapy	completed
NCT06504901	tyrosin kinase (PD-1)	Tislelizumab Interleukin-2 Capecitabine Oxaliplatin Neupogen	Clinical	30	Overcoming the limitations of single-agent immunotherapy through multifaceted immune modulation	not yet recruiting
NCT06280495	tyrosin kinase (PD-1)	Serpulimumab and bevacizumab	Clinical	156	Enhancing the immune microenvironment in the liver, increasing T lymphocyte infiltration, and consequently improving the post-op prognosis in resectable CRLM	recruiting
NCT06199232	tyrosin kinase (PD-1)	HAIC + targeted therapy + Tislelizumab	Clinical	47	Efficacy and safety of targeted treatment based on ctDNA genotyping as salvage treatment for advanced CRCLM failed from the standard systemic treatment	recruiting
NCT06590259	tyrosin kinase (PD-1)	Sintilimab + mFOLFOX6 or FOLFIRI + bevacizumab or cetuximab	Clinical	20	Evaluate the efficacy and safety of multi-mode ablation combined with systemic therapy	recruiting
NCT06794086	tyrosin kinase (PD-1)	PD-1 Monoclonal Antibody plus SBRT	Clinical	24	Improve the objective response rate (ORR), achieve better long-term survival benefits, and enhance quality of life in unresectable CRLM	recruiting
NCT06045286	tyrosin kinase (PD-1)	Radiation: High- and low-dose radiotherapy plus PD-1 inhibitors	Clinical	30	Investigate the efficacy and safety of failed second-line immunotherapy or above	recruiting

* Estimated or actual enrollment.

## Data Availability

The original contributions presented in this study are included in the article. Further inquiries can be directed to the corresponding author(s).

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
