# Peer review of "Liver Extracellular Matrix in Colorectal Liver Metastasis"

_cancers, 2025, doi:10.3390/cancers17060953_

Round 1
Reviewer 1 Report
Comments and Suggestions for Authors
The manuscript provides a comprehensive narrative review on the pivotal role of the hepatic extracellular matrix (ECM) in colorectal liver metastases (CRLM). The review effectively synthesizes existing knowledge, highlighting the complex interplay between ECM components, tumor progression, and chemoresistance. While the manuscript is thorough and informative, it requires attention to grammatical and typographical issues and the inclusion of additional references to strengthen some arguments. Below are detailed comments provided in the order they appear in the manuscript.
Abstract:
- Line 2: "Liver is the most common site of metastasis of colorectal cancer (CRC)..." should be corrected to "The liver is the most common site of metastasis from colorectal cancer (CRC)..."
- Line 4: "Tumor microenvironment, particularly the extracellular matrix (ECM), is also critical in the CRC metastasis process as well as chemoresistance." This sentence is slightly awkward. Consider rephrasing to: "The tumor microenvironment, particularly the extracellular matrix (ECM), plays a critical role in CRC metastasis and chemoresistance."
Introduction:
Line 45: "Colorectal cancer (CRC) is the third most frequent neoplastic disease worldwide and the second leading cause of cancer-related mortality..." Add citations to support this epidemiological data. Provide updated statistics on cancer in general as well as this cancer type prevalence, including survival rates, to highlight the critical need for prognostic biomarkers. Cite “Cancer statistics, 2024, 2024”. Then give intro in cancer therapy in general, cite NIH paper “Cancer treatments: Past, present, and future, 2024” (PMID: 38909530)for more information.
- Line 56: "Up to 25% of patients are already found to have synchronous liver metastases." This is a critical statistic that requires a citation.
- Line 67: "CRC cells can spread via the lymphatic system, though this is a less common route of liver involvement..." Add a citation to support this pathway of metastasis.
- Line 91: "Filling these gaps is a key-point, as a more complete understanding improving outcomes for patients with metastatic CRC." This sentence is awkward; consider revising to: "Addressing these gaps is crucial for improving outcomes in patients with metastatic CRC."
The Extracellular Matrix: Composition and Function in Healthy Tissue:
- Line 113: "ECM is a complex, dynamic network of more than 300 different molecules that provides scaffolding maintaining tissue structure and integrity..." Consider rephrasing to: "The ECM is a complex, dynamic network of over 300 molecules that provides scaffolding to maintain tissue structure and integrity..."
- Line 125: "Adhesive glycoproteins (including fibronectins and laminins, which mediate cell-ECM interactions..." Ensure proper citation for the functions of fibronectins and laminins.
Extracellular Matrix (ECM) in Cancer:
- Line 169: "ECM has also demonstrated to be crucial not only in supporting tumor cellular growth but also in driving malignant transformation." Rephrase to: "The ECM is crucial not only in supporting tumor cell growth but also in driving malignant transformation."
- Line 184: "Once tumor cells enter the bloodstream, ECM components assist in the establishment of secondary tumors by providing a supportive niche for metastatic colonization." Add a reference to support this statement.
- Discuss circRNAs function in ECM, cite “SP1-induced circ_0017552 modulates colon cancer cell proliferation and apoptosis via up-regulation of NET1”
Liver ECM in Regulating Cell Fate, and its Role in CRC Metastasis:
- Line 254: "Liver ECM is a highly specialized and dynamic, cross-linked network of macromolecules that plays a crucial role in maintaining hepatic ultrastructure and function." Consider simplifying: "The liver ECM is a specialized, dynamic network of macromolecules crucial for maintaining hepatic structure and function."
- Line 278: "Although the functional implications of these spatial variations remain unclear, they warrant additional exploration." This statement would benefit from a citation or example of ongoing research in this area.
ECM Remodeling in Normal and Pathological Liver Conditions:
- Line 357: "Under normal conditions, ECM turnover ensures a balanced deposition and removal of ECM components, enabling the liver to adapt to metabolic demands and repair minor injuries." Add a citation supporting ECM turnover's role in liver homeostasis.
- Line 368: "The interplay between cancer cells and the altered ECM creates a permissive environment for tumor progression and dissemination." Ensure this is supported by current literature.
ECM and Chemoresistance:
- Line 551: "The ECM plays a critical role in cancer chemoresistance, previously viewed as a passive scaffold but now recognized for actively promoting tumour resilience..." Simplify for clarity: "The ECM, once considered a passive scaffold, is now recognized for actively promoting tumor resilience and chemoresistance."
- Line 577: "The fibrotic and collagen-rich ECM, characteristic of CRLM, significantly increases tissue stiffness and interstitial pressure..." Add references that link ECM stiffness to chemoresistance in CRLM.
- Line 593: "ECM components such as fibronectin, laminin, and collagen engage integrin receptors on cancer cells, activating survival pathways." Add citations to support the involvement of these ECM components in integrin-mediated chemoresistance.
- Suggest potential future study for drug resistance target discovery, such as CRISPR screening, and how these can help, suggest to refer to “CRISPR screening and cell line IC50 data reveal novel key genes for trametinib resistance, 2024”
Therapeutic Implications and Future Directions:
- Line 693: "Targeting the ECM directly addresses its role in CRLM by disrupting pro-tumorigenic remodeling and improving drug delivery." Consider adding examples of current clinical trials or therapies that target ECM components in CRLM.
- Line 731: "Integrin inhibition has been shown to synergize with chemotherapeutics, increasing apoptosis and reducing tumor burden in preclinical CRLM models." Add a citation to support this therapeutic approach.
Author Response
you will find all the response in the attached file and the relative adjustment in the article highlihighlighted by the colour yellow

Reviewer 2 Report
Comments and Suggestions for Authors
The manuscript provides a comprehensive review of the role of the hepatic extracellular matrix (ECM) in colorectal liver metastases (CRLM), detailing its contributions to tumor progression, chemoresistance, and potential therapeutic targets. The discussion is well-structured, and the integration of clinical and basic research enhances its relevance. However, the manuscript would benefit from a more explicit differentiation between established findings and emerging hypotheses, as well as a clearer emphasis on the translational potential of ECM-targeted therapies. Some sections, particularly those discussing ECM remodeling and mechanotransduction, could be more concise to improve readability without compromising depth. Additionally, a summary table synthesizing key ECM components, their roles in CRLM, and potential therapeutic strategies would enhance clarity for readers. Minor language refinements are also suggested to ensure clarity and coherence. Overall, the review is insightful and well-researched, and with slight refinements, it will significantly contribute to the understanding of ECM dynamics in CRLM.
Author Response
you will find all the response in the attached file and the relative adjustment in the article highlihighlighted by the colour green

Reviewer 3 Report
Comments and Suggestions for Authors
This is a nice review summarizing the current understanding of the role played by the ECM in the development and progression of cancer metastasis, with a focus on liver ECM and colorectal cancer metastases. The review contains an exhaustive explanation of the pathophysiological processes involved; however, the link to clinical practice was not given proper attention. Section 8 (Therapeutic Implications and Future Directions) was expected to provide a guide on such evidence, but the authors missed this opportunity. My suggestion for the authors is to elaborate on the issue of therapeutic implications more thoroughly. This section would benefit from systematization, which could be introduced through the addition of a novel table describing existing research. The pre-clinical studies need to be distinguished from the clinical studies, and the authors could address both completed and ongoing studies on the topic by delving into one of the existing public domains, such as ClinicalTrials.gov.
More specific comments are as follows:
- Line 44. It is unclear why Introduction section has a single subheading. In my opinion, this subheading is redundant and could be removed.
- Lines 185-189 need to be referenced.
- The titles of Tables 1-4 need to be placed on the top, not the bottom.
- Line 352. It is not clear why this second subsection of Section 4 is numbered as "1". Basically, all subsections need to be numbered consequently, i.e., 4.1, 4.2, 4.3., etc.
- The review lacks "Conclusions" section.
Author Response
you will find all the response in the attached file and the relative adjustment in the article highlihighlighted by the colour blue

Round 2
Reviewer 1 Report
Comments and Suggestions for Authors
good
Comments on the Quality of English Languagegood
Reviewer 3 Report
Comments and Suggestions for Authors
Well done!